# Seismic Resistance and Parametric Study of Building under Control of Impulsive Semi-Active Mass Damper

**Ming-Hsiang Shih [1] and Wen-Pei Sung [2],***

[1] Department of Civil Engineering, National Chi-Nan University, Nantou 54561, Taiwan; mhshih@ncnu.edu.tw
[2] Department of Landscape Architecture, National Chin-Yi University of Technology, Taichung 41170, Taiwan
* Correspondence: wps@ncut.edu.tw; Tel.: +886-963-179-668

**Abstract:** When high-rise buildings are shaken due to external forces, the facilities of the building can be damaged. A Tuned Mass Damper (TMD) can resolve this issue, but the seismic resistance of TMD is exhausted due to the detuning effect. The Impulsive Semi-Active Mass Damper (ISAMD) is proposed with fast coupling and decoupling at the active joint between the mass and structure to overcome the detuning effect. The seismic proof effects of a high-rise building with TMD and ISAMD were compared. The numerical analysis results indicate that: (1) the reduction ratio of the maximum roof displacement response and the mean square root of the displacement reduction ratio of the building with the ISAMD were higher than 30% and 60%, respectively; (2) the sensitivity of the efficiency index to the frequency ratio of the ISAMD was very low, and detuning did not occur in the building with the ISAMD; (3) to achieve stable seismic resistance of the ISAMD, its frequency ratio should be between 2 and 4; (4) the amount of displacement of the control mass block of the ISAMD can be reduced by enhancing the stiffness of the auxiliary spring of the ISAMD; and (5) the proposed ISAMD has a stable control effect, regardless of the earthquake distance.

**Keywords:** Impulsive Semi-Active Mass Damper; frequency ratio; mass ratio; structural displacement; maximum roof displacement





## 1. Introduction

As science and technology advance further, the demand for urban development continues, so high-rise buildings and skyscrapers are constantly being built. Many important cities think of such buildings as landmarks. However, if such buildings are shaken by wind or earthquake forces, the users may experience both physical and psychological discomfort. Such shaking can also damage the facilities in the building, such as elevators, water towers and pipelines. Although these effects are rarely considered as design control factors for structural security, long-term dynamic deformation can shorten the fatigue lifetime of structural materials. Thus, the shock absorption of high-rise buildings is a very important issue.

Unlike energy dissipation technology [1–16], a TMD needs to be installed on the higher floors of a building to provide a fine shock absorption effect. The weight of a mass damper is much less than that of the structure, and the available space for such an installation is limited. However, the TMD is a popular passive control technology for super high-rise buildings. Currently, TMDs have been installed in buildings worldwide, such as the Citigroup Center, USA [17]; CN Tower, Canada [18]; John Hancock Tower, Boston, USA [19]; and Taipei 101, Taiwan [20]. The main purpose of a mass damper is to reduce wind-generated building vibrations by 30–40%. Many empirical formulas of the optimal parameters for the frequency ratio and damper ratio of a TMD [21–27] have been proposed over the past 20 years. The optimization of the design of a TMD is not complicated if the basic natural frequency is calibrated well. However, changes to the service modes of a building, aging materials and other problems can contribute to changes in the structural frequency of a building, and. over time, the frequency ratio of the TMD will deviate from

the effective frequency ratio interval. This deviation reduces the shock absorption effect of the TMD. Many studies have pointed out that, when the material or geometric conditions of a structure enter a nonlinear state under the action of a strong wind or seismic force, the shock absorption effect of a TMD is almost completely lost. The reason is that the basic natural frequency of the structure decreases and the frequency ratio of the TMD shifts. This phenomenon is called the "detuning effect" of a TMD. To mitigate or obviate this effect, many improvement strategies have been proposed by many scholars [28–44], including the Multi-Tuned Mass Damper (MTMD), Active Variable Interal-Semi-Active Mass Damper (AVI-SAMD) and Switched Semi-Active Mass Damper (Switch-type SAMD).

The natural frequency of the Semi-Active Mass Damper (SAMD) is similar to the natural frequency of the structure. Therefore, the natural frequency of the semi-active mass dampers with impulsive reaction [45] provides twice the natural frequency of the structure. Therefore, a kind damper was defined to distinguish the difference between SAMD and this new structural control mechanism, the Impulsive Semi-Active Mass Damper (ISAMD). The main purpose of ISAMD is to address the main defects of the TMD and thereby to improve the shock absorption effect. The ISAMD adopts the advantages of both the TMD and the Active Mass Damper (AMD) and requires only a limited power supply. The control method of the ISAMD involves simply locking and unlocking the connection between the structure and the control mass. These "Unlock" and "Lock" actions of the ISAMD are determined solely according to the structural reactions of the structure under excitation by external forces. Because this action changes the natural frequency of the entire structure, the structural reactions can be reduced. The research achievements of the impulse semi-active mass control mechanism [46] show that the optimal design frequency ratio for this ISMAD is around 4 with a mass ratio around 0.04–0.06 as well as suitable mass distribution at each floor. The frequency ratio of structure with ISAMD should be less than 4.0 to avoid enlarging the maximum acceleration responses. Regardless of the structural frequency misestimation ratio, the detuning phenomenon had little effect on structure under control of ISAMD. Nevertheless, a higher or lower misestimation ratio of the structure frequency caused a worse shock absorption effect of structure under control of TMD [47]. To overcome the defects of TMD, the control mechanism of the ISAMD and the Directional Active Joint of the ISAMD were developed in this research to derive the control law based on the control characteristics of the proposed ISAMD, and the Vector Form Instinct Finite Element method (VFIFE) was applied to perform a dynamic numerical simulation for a structure with TMD or ISAMD, respectively. Then, time histories of structural responses of high-rise buildings with a TMD and the ISAMD under excitation of various near-fault and far-field ground motion records were compared to determine the seismic proof capabilities and the maximum displacements of the control mass blocks of these two dampers under different parameters. In addition, the shock absorption effect and displacement of the control mass block of the ISAMD under various parameters with an auxiliary spring and damper are discussed in this paper, as is the seismic resistance of a building with a TMD or ISAMD affected by near-fault and far-field ground motion. The goal of this study was to develop this new ISAMD to reduce vibrations induced by wind and earthquake forces in high-rise buildings.

## 2. Concept of the ISAMD

The main idea of the ISAMD is that an active joint between a structure and a mass damper can be locked or unlocked to form a series, namely a structure–spring–active joint–mass damper, as shown in Figure 1.

The components of the proposed ISAMD are as follows: (A) control mass block; (B) active component unit (comprising (B1) switching spring and (B2) active joint); (C) vibration sensing unit (comprising (C1) structural vibration sensors and (C2) vibration sensors for the control mass block); (D) controller; and (E) fixed component unit (comprising (E1) fixed rebound device and (E2) energy dissipation device). The workflow of the proposed ISAMD is as follows:

(1) Motion of the structural floor and the control mass block is detected by motion sensors (accelerometers, C1 and C2). The signal is actively retrieved by the controller.
(2) The controller converts the acceleration signal to velocity and displacement signals.
(3) The controller determines whether to "Unlock" or "Lock" the active joint according to the control law and outputs the control signal to the active joint.
(4) The active joint (B2) releases or captures the switching spring (B1) according to the control signal from the controller.
(5) Steps (1)–(5) are repeated.

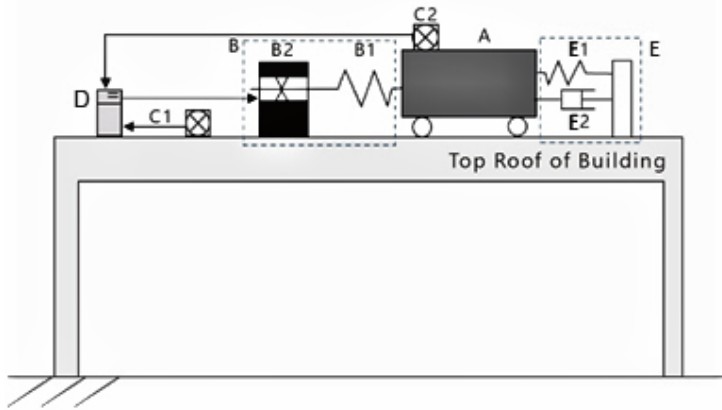

**Figure 1.** Composition of Impulsive Semi-Active Mass Damper.

The control can perform on/off switching by using a quick-reaction electromagnetic server or solenoid valve. In the "Unlock" status of the active joint, there may be no interaction between the control mass block and the structure, or a weak spring (E1) and energy dissipation device (E2) may maintain a weak interaction force to reduce the stroke of the control mass block. In the "Lock" status of the active joint, the control mass block and the main structure have a strong connection. The interaction force induced by the deformation of the strong spring is applied to change the movement behaviors of the main structure and the control mass block. The timing of the switching of the active joint can be achieved with positive/negative symbols of the work done by the control mass block on the main structure, as follows:

(1) Lock to Unlock: When the inner force of the switching spring (B1) begins to do positive work on the structure, the ISAMD switches to the "Unlock" status and the structure and the control mass become separated.
(2) Unlock to Lock: When the ISAMD switches to the "Lock" status, the switching spring (B1) performs negative work on the structure. Then, the controller (D) sends a "Lock" command to the active joint (B2) to reconnect the main structure and control mass with the switching spring.

## 3. Developing Directional Joint of the ISAMD

The process from Unlock to Re-Lock of structural control device can cause time delays due to equipment problems. The longer the time delay lasts, the less control it will have, which can seriously affect the control effect of control device. To reduce the defects of time delay problems and achieve the above Unlock/Lock switching process, a directional active joint is developed in this research. The main components are as follows: (A) bevel casting tube (blue); (B) locking steel ball (green); (C) switching tube (red); (D) push rod (yellow); (E) end cover plate (grey); (F) central slider (white); and (G) spring (black) (Figure 2). An exploded diagram of the directional active joint is provided in Figure 3. When a force on the central slider pushes it to the right, the steel ball is in contact with the central sliding rod and the double bevel casing. It is automatically locked and unable to move to the right. The steel balls on the left maintain contact with the central sliding rod and double bevel

casing. Conversely, when the central slider slides to the left under force, the normal force of the steel balls on the left side disappears. Since there is no locking power, the sliding rods are free to move to the right. If the steel balls on the right (Figure 2) also maintain contact with the central sliding rod and the double bevel casing, the contact provides a restraint function opposite to that of the steel balls on the left side. Thus, the slider can slide to the right without moving to the left.

Therefore, this joint, which can optionally lock the movement in one direction, is installed in the ISAMD to accurately and automatically combine the control mass block and the main structure for a short impact and then separate them. In addition, this directional active joint can avoid delaying the re-lock action and allows easy simplification of the control law. It greatly increases the reliability of the control mechanism. The directional joint and the control mass block pass through the spring–damper to connect with the top of the structure at both ends of the sliding shaft, as shown in Figure 4. The main hardware of the ISAMD is shown in Figure 5.

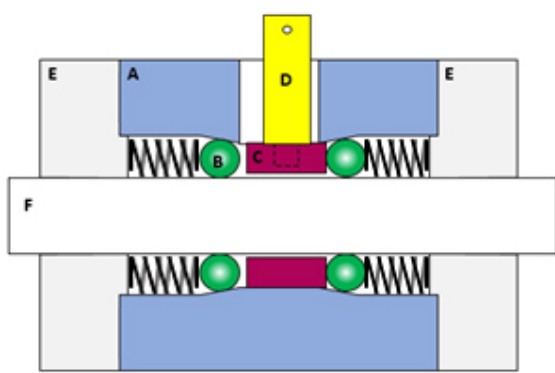

**Figure 2.** Section of directional active joint (schematic view).

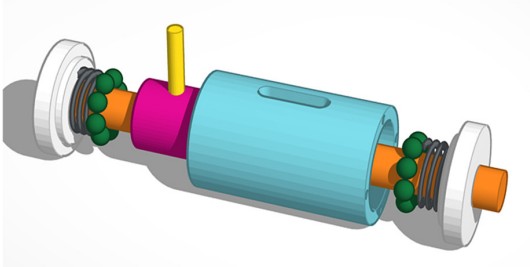

**Figure 3.** Exploded diagram of the directional active joint.

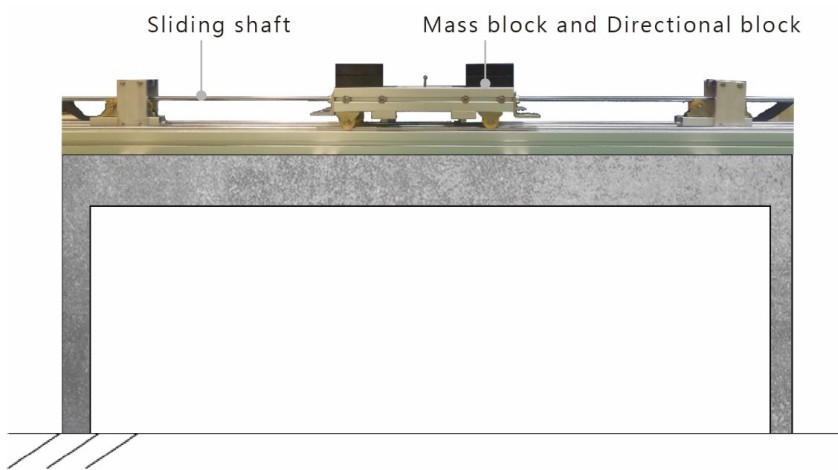

**Figure 4.** The main hardware of the ISAMD with the directional joint, sliding car and mass block.

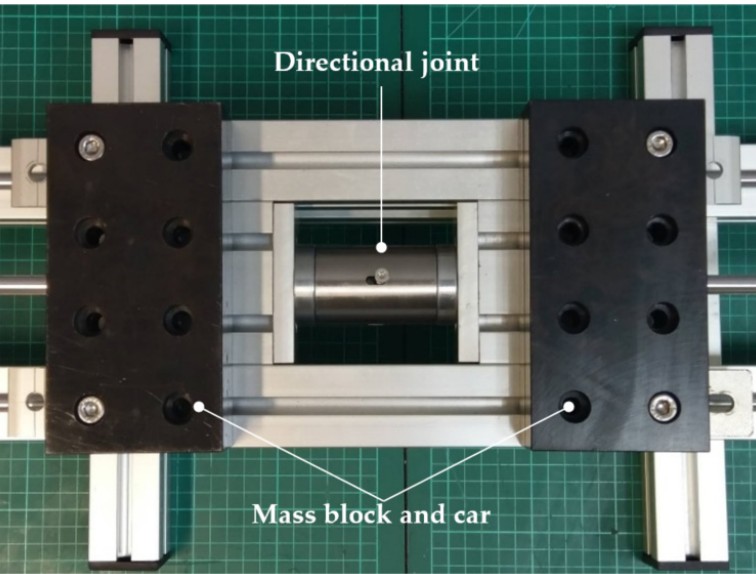

**Figure 5.** Combination of directional joint, sliding car and mass block.

## 4. Analysis Model

### 4.1. Control Law of the ISAMD

The control law of the ISAMD is based on the switching spring, which can capture the structure at the right time for the control mass block to absorb the kinetic energy of the structure. When the control mass block begins to do positive work on the structure, the switching spring is released to maximize the negative work to achieve the maximum energy dissipation effect. The basic condition for the timing is that the mass control block can do negative work on the structure. To maximize this negative work, it is also necessary to consider the structural velocity and acceleration responses. The proposed control law in this study is as follows:

$$\text{Status of Joint is "Unlock", When } (W_V V_s - W_A A_s) \times V_{CS} \leq 0, \text{ Switch to "Lock"}$$
$$\text{Status of Joint is "Lock", When } V_S \times D_{CS} > 0, \text{ Switch to "Unlock"} \tag{1}$$

where

$V_S$ is the moving velocity of the controlled structure, where the rightward direction is positive;

$V_{CS}$ is the moving velocity of the control mass block relative to that of the controlled structure, where the rightward direction is positive; and

$D_{CS}$ is the displacement of the control mass block relative to that of the controlled structure when the active joint is locked. It is equal to the deformation of the switching spring.

When $V_S \times D_{CS} > 0$, the ISAMD does positive work on the structure. $W_V$ is the weighting of velocity, which is greater than or equal to 0. $W_A$ is the weighting of acceleration, which is greater than or equal to 0. The flowchart of the control law of the ISAMD is shown in Figure 6.

According to this control law, the lock/unlock time of the active joint must take into account the structural acceleration. In fact, the sensor detects the trend of velocity changes in its motion. Therefore, the acceleration sensor is not required as long as the first difference of the speed sensing value of the controller unit or the second difference of the displacement sensing value of the controller unit obtains acceleration responses.

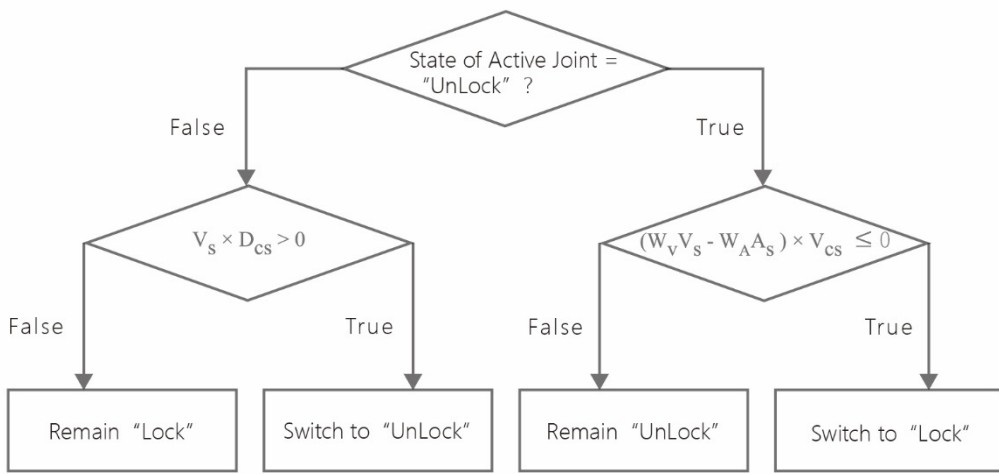

**Figure 6.** Control Low flowchart of the ISAMD.

*4.2. One-Dimensional Vector Form Instinct Finite Element Method, VFIFE*

During the action of the ISAMD, the directional joint of the ISAMD is applied to lock or unlock the connection between the ISAMD and the structure. Thus, the Vector Form Instinct Finite Element Method (VFIFE) is applied to analyze the structural responses of the structure with the ISAMD. The steps of the analysis are as follows:

Step 1: Discretization

The VFIFE method is used to discretize the joints of the nodes and elements, where the nodes are the vertices of the element. Nodes are assigned to the mass of the element and become the contact point between two separate elements. The element provides internal force based on the overall coordinates of its nodes and is assigned to the node. The equation is as follows:

$$\vec{F}_{in,j}^{k}(t_i) = \vec{f}\left(\vec{u}(t_i), \dot{\vec{u}}(t_i), \Omega^k\right) \tag{2}$$

where
$\vec{F}_{in,j}^{k}(t_i)$ is the internal force vectors of the $k$ element at time point $t_i$ to maintain this element at the time the node coordinate state is in the $j_{th}$ node. This internal force is affected by the equation of motion, which includes mass matrix, damping matrix and stiffness matrix.
$\vec{u}(t_i)$ and $\dot{\vec{u}}(t_i)$ are the system displacement vector and velocity vector at time point $t_i$, respectively;
$\Omega^k$ represents the properties of the $k_{th}$ element, including the material properties and geometric definitions.

Step 2: Force Equilibrium

At each time point $t_i$, the internal forces acting on each node associated with the element can be calculated by the node displacement and velocity vector at that time. The external and internal forces on the degrees of freedom of all nodes can be accumulated to achieve the unbalanced force vector on each node's degree of freedom. The equation can be expressed as follows:

$$\Delta \vec{F}_j(t_i) = \vec{F}_{ext,j}(t_i) - \sum_{k=1}^{n_e} \vec{F}_{in,j}^{k}(t_i) \tag{3}$$

where
$\Delta \vec{F}_j(t_i)$ is the balance force vector of the $j_{th}$ node at time point $t_i$ and
$\vec{F}_{ext,j}(t_i)$ is the external force vector of the $j_{th}$ node at time point $t_i$.

Step 3: Predict displacement vector at the next time step $t_{i+1}$

The centered difference scheme is applied to explicitly estimate the displacement reaction at time point $t_{i+1}$ under the condition of acceleration responses as a known condition at time point $t_i$:

$$\ddot{\vec{u}}_j(t_i) = \frac{\vec{u}_j(t_{i+1}) - 2\vec{u}_j(t_i) + \vec{u}_j(t_{i-1})}{\Delta t^2} \tag{4}$$

where
$\ddot{\vec{u}}_j(t_i)$ is the acceleration vector of the $j_{th}$ node at time point $t_i$ and
$\Delta t$ is an analytical stride.

Then, the displacement reaction at time point $t_{i+1}$ can be expressed as follows:

$$\vec{u}_j(t_{i+1}) = M_j^{-1}\Delta\vec{F}_j(t_i)\Delta t^2 + 2\vec{u}_j(t_i) - \vec{u}_j(t_{i-1}) \tag{5}$$

where $M_j$ is the mass matrix of the $j_{th}$ node. It is a diagonal matrix.

Step 4: Repeat the above steps to complete the analysis.

## 5. Setting of the Analysis Model for the Shock Absorption Effect of a 10-Story Building with the ISAMD

To explore the reaction to seismic forces of a structure with multiple degrees of freedom and the ISAMD and define the effective interval of the control parameters of the ISAMD for control system design, a 10-story shearing building under excitation of various earthquake records was investigated. These numerical simulations compared the displacement reaction at the roof of the building and at the control mass block for a 10-story shearing building under control of a TMD and the ISAMD and under excitation of different earthquake records. Thus, the structural displacement and displacement reduction ratio of the control mass block under various combinations of parameters were compared to obtain the optimal design parameters. An analysis program with a VFIFE function was applied to perform numerical simulations. In those simulations, the element of the linear spring-damper in parallel was applied to simulate the elastic recovery force and damping provided by the column of the shear building as well as simulate the spring force and damping force between the TMD and the top floor of the main structure. The spring stiffness and weighting of the velocity and acceleration of the control law were set for the ISAMD. Time delay effects were not considered in this research.

### 5.1. Analysis Setting

The analyzed control subject matter in this study was the 10-story shearing building shown in Figure 7. To compare the control characteristics of the building under control of the ISAMD and TMD, this research analyzed the structural responses of the building with these two dampers under excitation of different earthquake records. The main parameters of the analysis included the parameters of the main structure and the control parameters of the ISAMD and TMD, described as follows.

#### 5.1.1. Main Structure Parameters

The bare structure was the 10-story shearing building. Assuming a mass $m_s$ of each floor of 500 tons, the story stiffness $k_s$ and damper $c_s$ were 883,645 kN·m and 2813 kN·s/m, respectively. The first modal frequency and damper ratio were 1.0 Hz and 0.01, respectively.

#### 5.1.2. Control Parameters of the ISAMD

The control parameters of the ISAMD can be divided into two categories: (1) system hardware control parameters, including the control mass block ratio ($\mu$), frequency ratio of the control mass block ($\gamma_f$) and the damping ratio of the control mass block ($\xi_a$); and (2) control law parameters, including the weight of velocity response ($w_v$) and acceleration response ($w_a$) of the control law.

The control mass block ratio can be defined by the rate of the mass of the control block and structure:

$$\mu = \frac{m_a}{\sum m_s} \tag{6}$$

Then, the frequency ratio of the control mass block can be defined as the rate of a natural vibration frequency and the first modal frequency of the main structures:

$$\gamma_f = \frac{f_a}{f_0} = \frac{\sqrt{k_a/m_a}}{2\pi f_0} \tag{7}$$

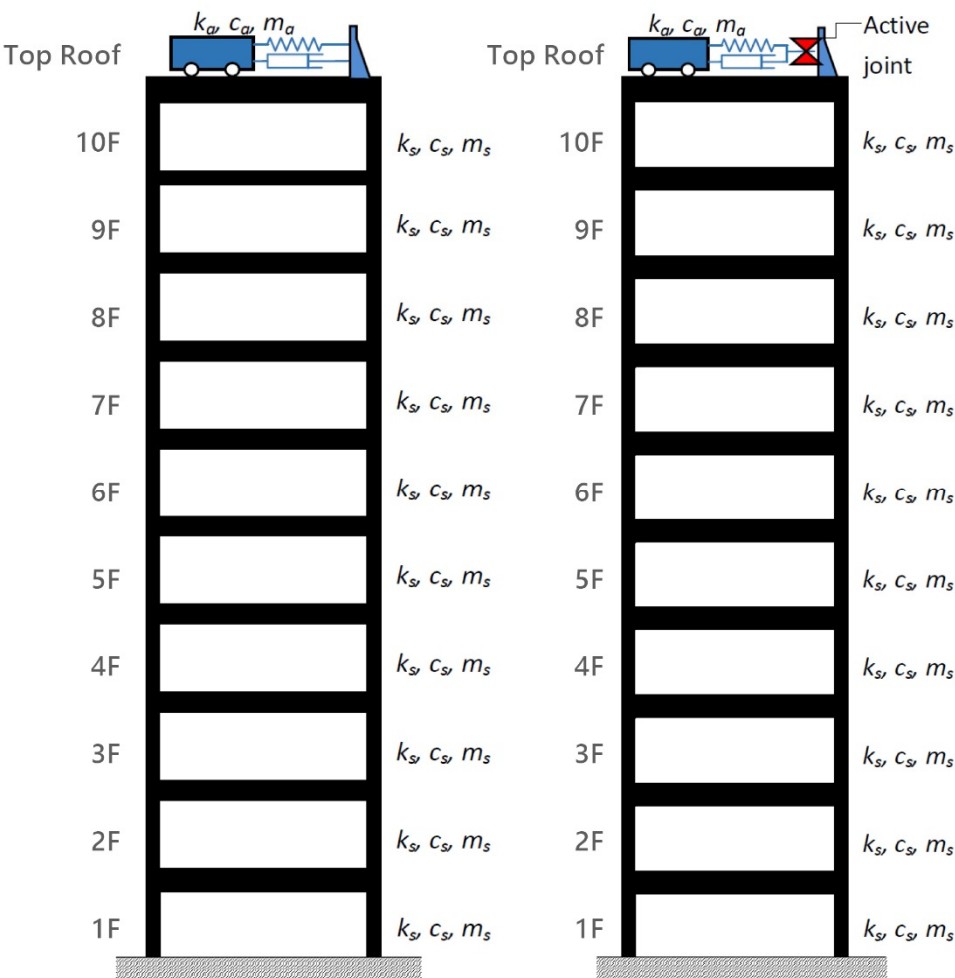

(**a**) Building under control of the TMD　　　　　　(**b**) Building under control of the ISAMD

**Figure 7.** Analysis model of Building with TMD and ISAMD.

The weight of velocity response ($w_v$) and acceleration response ($w_a$) of the ISAMD control law affect the timing of the "releasing" and "capturing" status of the control mass block and the main structure. If $w_v : w_a = 1 : 0$, the "releasing" (unlock) or "capturing" (lock) status is determined by the structural velocity responses at the installation position of the control mass block. The timing of the combination of the ISAMD and structure is at the point when the direction of the structural motion reverses, which is the maximum or minimum structural displacement, at which point the structural velocity is zero. In contrast, when $w_v : w_a = 0 : 1$, the "releasing" or "capturing" status is determined by the structural acceleration responses at the installation position of the control mass block. The timing of the combination of the ISAMD and the structure is at the point when the

direction of the structural acceleration reverses, i.e. the structural displacement is zero and the structural acceleration reaches the maximum or minimum acceleration. The ratio between these two extreme ratios causes the switching time point to be between these two extreme time points, at which point the velocity and acceleration direction are the same. Five different proportions are compared to determine the seismic resistance in this study. The control parameters of the ISAMD are listed in detail in Table 1.

**Table 1.** Analysis setting value [47].

| Analysis Parameter Category | Parameter Range |
|---|---|
| Parameters of main structure | Structure type: 10DOF shear building<br>Story mass: $m_a$ = 500 tn<br>Inter-story stiffness: $k_a$ = 883,645 kN/m<br>Inter-story damping: $c_a$ = 2813 kN·s/m<br>First modal frequency: $f_0$ = 1.0 Hz<br>First modal damping ratio: $\xi_0$ = 0.01 |
| Control Parameters of the ISAMD | Mass ratio: $\mu$ = 0.02<br>Freq. ratio: $\gamma_f$ = 0.8145~6.0 @rate = 0.95<br>Damping ratio: $\xi_a$ = 0.01~0.10<br>Weight ratio:<br>$w_v/w_a = 1/0, 4\pi f_0/0.414, 2\pi f_0/1, 2\pi f_0/2.414, 0/1$ |
| Control Parameters of the TMD | Mass ratio: $\mu$ = 0.02<br>Freq. ratio: $\gamma_f$ = 0.8145~1.2277@rate = 0.95<br>Detuning rate: $\Gamma$ = 18.55%~22.77%<br>Damping ratio: $\xi_a$ = 0.071~0.10 |
| Seismic waves | 26 records of ground acceleration of earthquakes with epicentral distance = 1–339 km (see Table 2) |

5.1.3. Control Parameters of the TMD

The control parameters of the TMD are as follows. The control mass block ratio ($\mu$), frequency ratio ($\gamma_f$) and damper ratio ($\xi_a$) have the same definitions as those of the ISAMD. To compare the influence of the shock absorption ratio by the detuning effect, the frequency ratio ($\gamma_f$) of the control mass block of the TMD was changed. The range of variation of this frequency ratio was around 0.8145–1.2277. The detuning frequency ($\Gamma$) is the ratio of the difference of the frequency ratio of the control mass block and the optimal frequency ratio of the TMD to the optimal frequency ratio of the TMD:

$$\Gamma = \frac{\gamma_f - \gamma_{f,opt}}{\gamma_{f,opt}} \times 100\% \tag{8}$$

The control parameters of the TMD are listed in detail in Table 1 and were based on the formula of the optimal control parameters suggested by Lin et al. [48] to estimate the optimal frequency ratio and the damping ratio, 0.9694 and 0.07035, respectively. Therefore, the optimal frequency of the control mass block was 6.283 rad./s.

$$r_f = \frac{f_a}{f_p} = \left(\frac{a}{1+\mu}\right)^b ; \ a = 1.0 - \frac{\xi_p}{4}; \ b = 1.35e^{3.2\xi_p}; \ \xi_a = 0.46\mu^{0.48} \tag{9}$$

5.1.4. Earthquake Records

The seismic action of a near fault is short in duration. Thus, the maximum structural displacement occurs under a 1.5–2.5 cycle action of the main shock wave. The shock absorption effect of the TMD comes from the kinetic energy of the control mass block. The TMD does not provide enough energy to resist the structural movement in the early stage of an earthquake. Therefore, there is no noteworthy shock absorption effect of the TMD on the maximum structural displacement reaction under the action of a near-fault

ground motion. Conversely, the structural displacement reaction is controlled by the component of the resonant frequency under the action of a far-field ground motion. The maximum displacement reaction occurs at 3–5 reciprocating cycles because the control mass block has enough time to accumulate sufficient kinetic energy to confer seismic resistance. Therefore, the TMD and ISAMD have significant control effects on the maximum structural displacement reaction for far-field ground motions because the shock absorption effect of the structural control method is closely related to the actual time history of the earthquake force. To compare the control efficiencies of different control methods, diachronic analysis of a building under control of the TMD and ISAMD is necessary to analyze and compare the average ratio and standard deviation of the shock absorption effect. Records of 26 earthquakes with epicentral distances of 1–339 km were used to analyze the seismic resistance. Each earthquake's name, occurrence time, recording station, epicentral distance, seismic direction and original peak ground acceleration are listed in Table 2.

**Table 2.** Earthquake records for analytical use.

| No. | Earthquake | Year | Station | Epi. Dist. (km) | Dir. | PGA (g) |
|---|---|---|---|---|---|---|
| 1 | Kobe, Japan | 1995 | KJMA | 1 | NS | 8.21 |
| 2 | Northridge, USA | 1994 | Tarzana | 4 | EW | 17.45 |
| 3 | Santa Barbara, USA | 1978 | UCSB Goleta | 14 | NS | 3.40 |
| 4 | Chi-Chi, Taiwan | 1999 | TCU075 | 18 | NS | 2.57 |
| 5 | Northridge, USA | 1994 | Newhall | 19 | EW | 5.72 |
| 6 | Northridge, USA | 1994 | S_Monica | 23 | EW | 8.66 |
| 7 | Norcialtaly, Italy | 2016 | St_Angelo | 28 | - | 2.06 |
| 8 | Northridge, USA | 1994 | LA | 38 | EW | 3.48 |
| 9 | Northridge, USA | 1994 | SF bay | 40 | - | 1.64 |
| 10 | Norcialtaly, Italy | 2016 | Colfiolito | 40 | - | 2.55 |
| 11 | Loma Prieta, USA | 1989 | S_Cruz | 48 | NS | 3.62 |
| 12 | Calexico, USA | 2010 | Sierra | 77 | EW | 4.91 |
| 13 | Chile | 2010 | CCSP | 109 | EW | 5.94 |
| 14 | Sumatra, Indonesia | 2007 | PSKI | 125 | EW | 1.24 |
| 15 | Chile | 2016 | LL06 | 136 | - | 2.24 |
| 16 | Chile | 2010 | VA03 | 168 | - | 2.68 |
| 17 | Chile | 2010 | GO01 | 170 | EW | 2.32 |
| 18 | Chile | 2010 | CUR | 170 | EW | 4.66 |
| 19 | Chile | 2010 | GO04 | 175 | EW | 2.34 |
| 20 | Chile | 1980 | ANGO | 209 | NS | 6.84 |
| 21 | Chile | 1980 | MAT | 230 | NS | 3.37 |
| 22 | Alaska, USA | 2016 | HNE | 254 | EW | 2.07 |
| 23 | Chile | 1980 | LLO | 274 | NS | 3.19 |
| 24 | Chile | 2010 | S_Jose | 333 | NS | 4.61 |
| 25 | Chile | 2010 | S_Lucia | 334 | NS | 2.39 |
| 26 | Chile | 2010 | ColegioLasAmericas | 339 | NS | 3.02 |

### 5.2. Indices of Control Performance

The shock absorption effects of these two dampers were compared in the building with TMD or ISAMD control under excitation of near-fault and far-field ground motion records. Six indices of control performance were defined to investigate the seismic resistance.

(1)　Average ratio of the maximum roof displacement reaction ($J_1$):

$$J_1 = average\left(\frac{max(|Roof\ displacement\ with\ ISAMD\ or\ TMD|)}{max(|Roof\ displacement\ withoutcontrol|)}\right) \quad (10)$$

(2)　Standard deviation of the maximum roof displacement reaction ratio ($J_2$):

$$J_2 = stdev\left(\frac{max(|Roof\ displacement\ with\ ISAMD\ or\ TMD|)}{max(|Roof\ displacement\ withoutcontrol|)}\right) \quad (11)$$

(3)　Average Root Mean Square, RMS ratio of the roof displacement reaction ($J_3$):

$$J_3 = average\left(\frac{rms(Roof\ displacement\ with\ ISAMD\ or\ TMD)}{rms(Roof\ displacement\ withoutcontrol)}\right) \quad (12)$$

(4)　Standard deviation of the average RMS ratio of the roof displacement reaction ($J_4$):

$$J_4 = stdev\left(\frac{rms(Roof\ displacement\ with\ ISAMD\ or\ TMD)}{rms(Roof\ displacement\ withoutcontrol)}\right) \quad (13)$$

(5)　Average ratio of the roof maximum absolute acceleration reaction ($J_5$)

$$J_5 = average\left(\frac{max(Roof\ acceleration\ with\ ISAMD\ or\ TMD)}{max(Roof\ acceleration\ withoutcontrol)}\right) \quad (14)$$

(6)　Average ratio of the maximum displacement of the control mass block ($J_6$):

$$J_6 = average\left(\frac{max(|Mass\ displacement\ with\ ISAMD|)}{max(|Mass\ displacement\ with\ TMD|)}\right) \quad (15)$$

All indices of control performance in this research follow the axiom of the Smaller the Better (STB).

## 6. Results and Discussion

### 6.1. Analysis Results

The analysis results of the control performance indices of the building with TMD or ISAMD control under excitation of 26 near-fault and far-field ground motion records are shown in Figures 8–13. The optimal control performance indices of the TMD and ISAMD are listed in Table 3. Table 3 reveals that the optimal control performance indices of the building under control of the TMD or ISAMD were relative to the frequency ratio of the control mass block $\gamma_f$. The ISAMD, without additional auxiliary dampers or springs and only one spring and active joint, switched the status of "Unlock" to "Lock".

**Table 3.** The optimal control performance indices for the building with TMD or ISAMD under excitation of 26 seismic records.

| Parameter | | J1 | J3 | J5 | J6 |
|---|---|---|---|---|---|
| TMD | Value | 0.72 | 0.48 | 0.88 | N.A. |
| | min. at *rf* | 0.95 | 0.95 | 0.95 | N.A. |
| ISAMD $W_V : W_A = 1:0$ | Value | 0.65 | 0.40 | 0.85 | 5.43 |
| | min. at *rf* | 1.59 | 1.43 | 1.67 | 2.19 |
| ISAMD $W_V : W_A = \omega_0 : \sqrt{2} - 1:$ | Value | 0.62 | 0.38 | 0.91 | 3.79 |
| | min. at *rf* | 2.1 | 2.1 | 1.45 | 5.78 |
| ISAMD $W_V : W_A = \omega_0 : 1$ | Value | 0.65 | 0.38 | 0.91 | 3.87 |
| | min. at *rf* | 2.79 | 2.27 | 1 | 4.90 |
| ISAMD $W_V : W_A = \omega_0 : \sqrt{2} + 1:$ | Value | 0.67 | 0.38 | 0.93 | 3.98 |
| | min. at *rf* | 2.79 | 2.90 | 1.2 | 2.12 |
| ISAMD $W_V : W_A = 0:1$ | Value | 0.68 | 0.4 | 0.94 | 3.71 |
| | min. at *rf* | 3.25 | 3.25 | 1.36 | 3.42 |

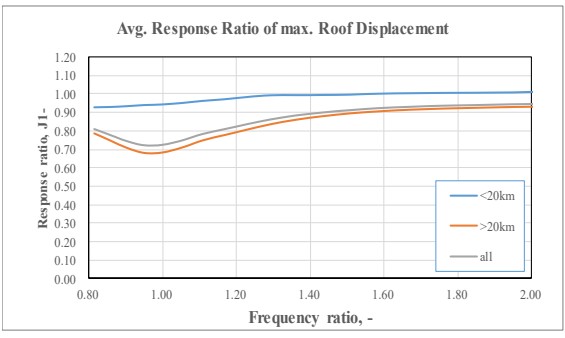

(**a**) Average response ratio of maximum roof displacement

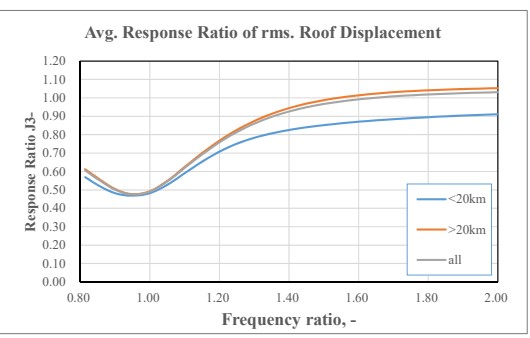

(**b**) Average response ratio of RMS roof displacement

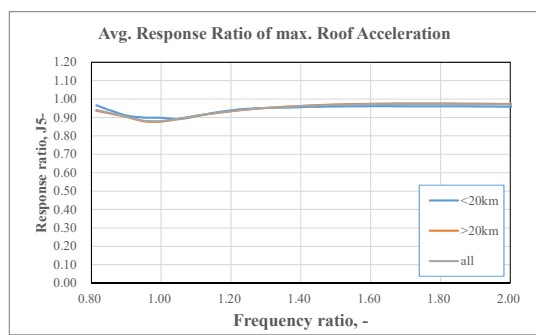

(**c**) Average response ratio of maximum roof acceleration

**Figure 8.** Response ratio of Structure with TMD under 26 seismic excitations [45].

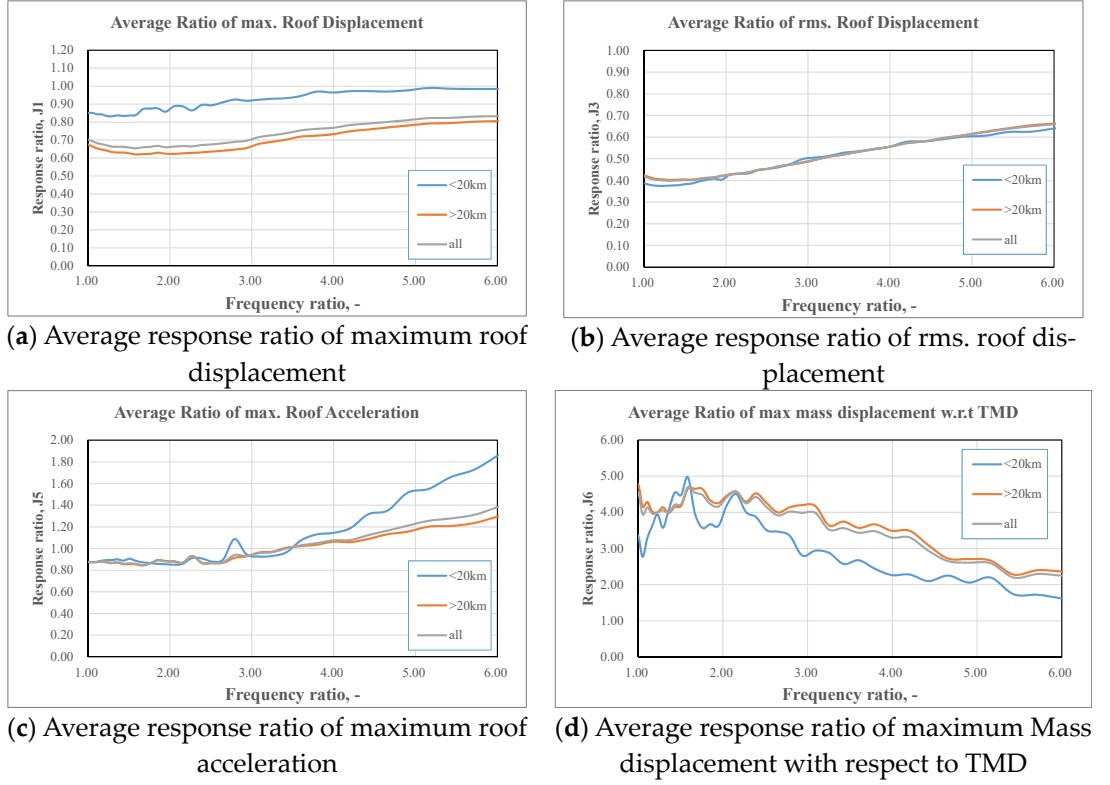

(**a**) Average response ratio of maximum roof displacement

(**b**) Average response ratio of rms. roof displacement

(**c**) Average response ratio of maximum roof acceleration

(**d**) Average response ratio of maximum Mass displacement with respect to TMD

**Figure 9.** Response ratio of Structure with the ISAMD with $W_V:W_A = 1:0$, without auxiliary damper and spring under 26 seismic excitations.

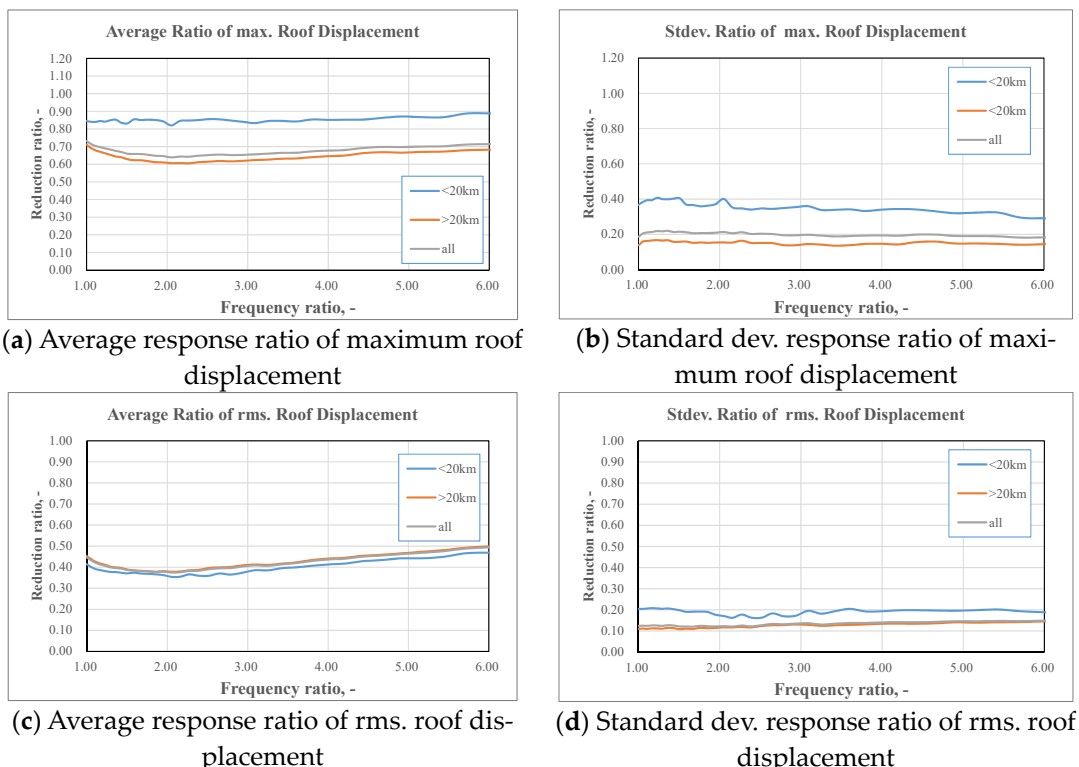

(**a**) Average response ratio of maximum roof displacement

(**b**) Standard dev. response ratio of maximum roof displacement

(**c**) Average response ratio of rms. roof displacement

(**d**) Standard dev. response ratio of rms. roof displacement

**Figure 10.** *Cont.*

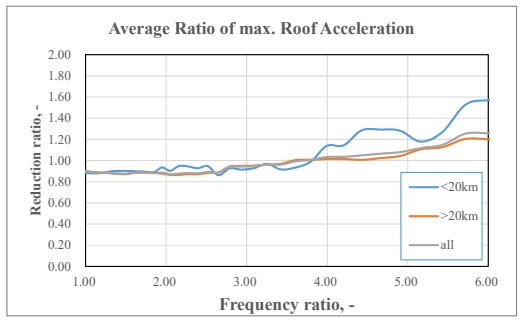

(**e**) Average response ratio of maximum roof acceleration

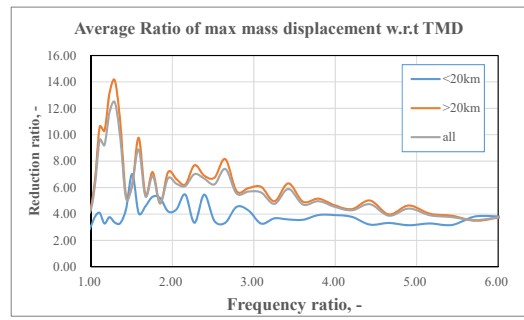

(**f**) Average response ratio of max Mass displacement with respect to TMD

**Figure 10.** Response ratio of Structure with the ISAMD with $W_V:W_A = \omega_0 : \sqrt{2} - 1$, without auxiliary damper and spring under 26 seismic excitations.

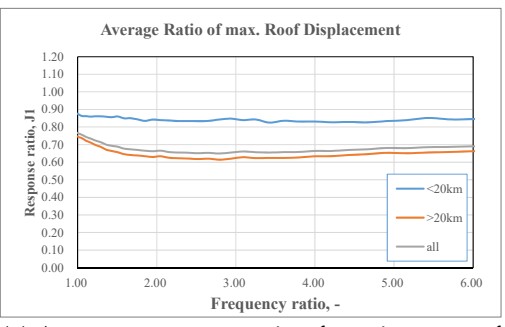

(**a**) Average response ratio of maximum roof displacement

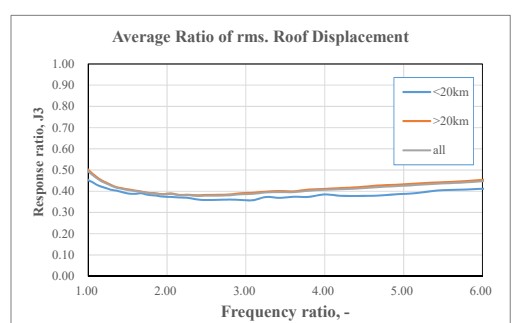

(**b**) Average response ratio of rms. roof displacement

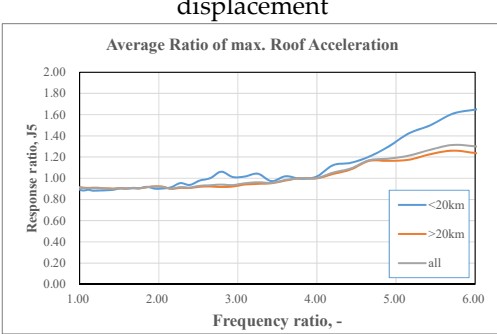

(**c**) Average response ratio of maximum roof acceleration

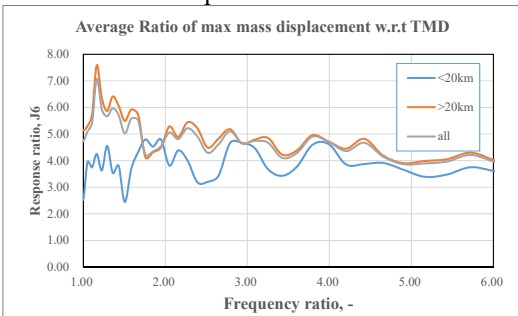

(**d**) Average response ratio of max Mass displacement with respect to TMD

**Figure 11.** Response ratio of Structure with the ISAMD with $W_V:W_A = \omega_0 : 1$, without auxiliary damper and spring under 26 seismic excitations [45].

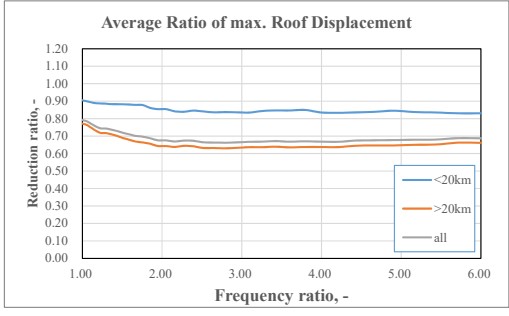

(**a**) Average response ratio of maximum roof displacement

(**b**) Standard deviation response ratio of maximum roof displacement

**Figure 12.** *Cont.*

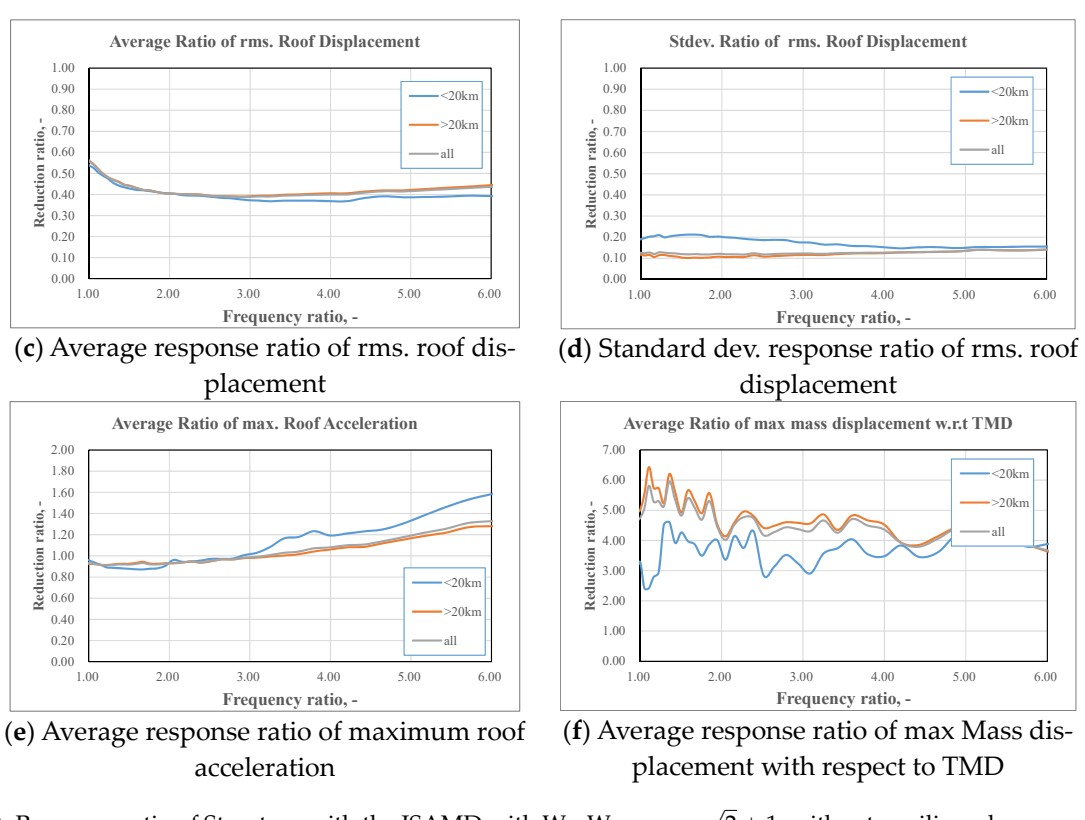

(**c**) Average response ratio of rms. roof displacement

(**d**) Standard dev. response ratio of rms. roof displacement

(**e**) Average response ratio of maximum roof acceleration

(**f**) Average response ratio of max Mass displacement with respect to TMD

**Figure 12.** Response ratio of Structure with the ISAMD with $W_V:W_A = \omega_0 : \sqrt{2}+1$, without auxiliary damper and spring under 26 seismic excitations.

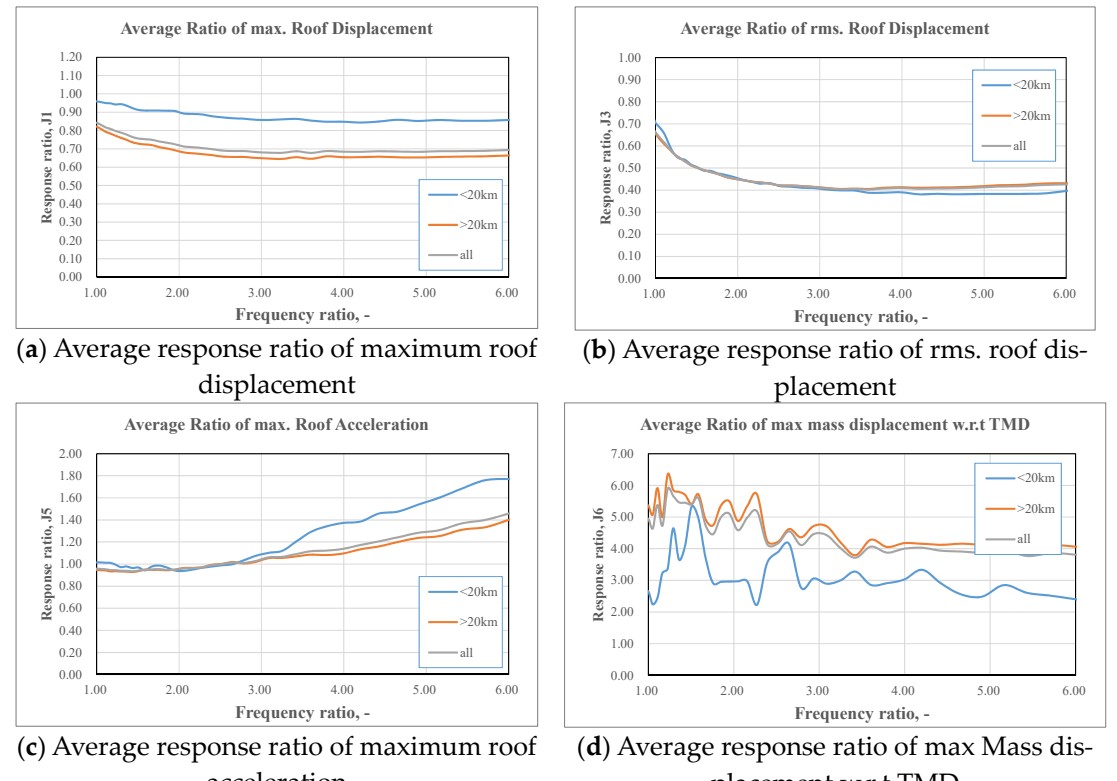

(**a**) Average response ratio of maximum roof displacement

(**b**) Average response ratio of rms. roof displacement

(**c**) Average response ratio of maximum roof acceleration

(**d**) Average response ratio of max Mass displacement w.r.t TMD

**Figure 13.** Response ratio of Structure with the ISAMD with $W_V:W_A = 0:1$, without auxiliary damper and spring under 26 seismic excitations.

*6.2. Discussion*

J1, as listed in Table 3, reveals that the maximum displacement reduction effect of the ISAMD was better than that of the TMD. The maximum displacement responses of the roof were further reduced by about 4–10%. The root mean square of the displacement reduction effect of the ISAMD was greater than that of the TMD, leading to a reduction of 8–10%. Conversely, the shock absorption effect of the structural acceleration responses of the building with the ISAMD was no better than that of the TMD. However, the maximum displacement response of the control mass block was significantly less than that of the TMD, so the required installation space of the ISAMD is 2–4 times that of the TMD. The comparison between the structural responses of the building with the TMD and the bare structure shown in Table 3 and Figure 6 indicates that the frequency ratio of the optimal control performance indices of the building with TMD was around 0.95. When the frequency ratio was slightly offset, the control performance indices J1, J3 and J5 increased. For example, when the frequency ratio was 1.2, J1, J3 and J5 were 0.83, 0.79 and 0.94, respectively. The shock absorption effect was very limited, manifesting the so-called detuning effect. In contrast, the relationship of the control performance indices to the frequency ratio, as presented in Table 2 and Figures 6–11, revealed that the minimum values of J1 and J3 occurred on a very flat curve. The frequency ratio was almost constant in this range. That is, the sensitivities of the control performance and frequency ratio were extremely low. Basically, this phenomenon could be expected because the maximum roof displacement reaction and the root mean square displacement of the building with the ISAMD had reduction rates of more than 30% and 60%, respectively. The control mass block displacement of the ISAMD was 2–4 times that of the TMD, as shown in Figures 9d, 10f, 11d, 12f and 13d. This suggests that the installation space of the ISAMD must be large. In fact, the higher ISAMD frequency of the control mass could provide the benefit of reducing the installation space, such as the simple pendulum TMD of the Taipei 101 building. It has a 6.8 s cycle, which requires a pendulum length of 11.48 m, occupying a space of four stories. Conversely, if the frequency ratio of the ISAMD is 3 with a 2.26 s cycle, the pendulum length only needs to be 1.27 m. It can be set up in one story. Therefore, the space requirement of the ISAMD may not be greater than that of the TMD.

6.2.1. The Maximum Roof Displacement and Root Mean Square Displacement of the Structure

The maximum roof displacement and root mean square displacement of the TMD under the condition of the optimal frequency ratio and those of the ISAMD with the weight ratio $W_V:W_A$ = 1:0 without a parallel auxiliary spring or damper under excitation of 26 earthquake records are compared in Figures 14 and 15. These two figures show that the shock absorption ratio of the ISAMD was 10% higher than that of the TMD. The shock absorption effect of the TMD varied greatly with the seismic load, and the standard deviation was about 0.26 and 0.17. The standard deviations of the ISAMD were only 0.20 and 0.10. In other words, the reliability of the displacement reaction of the control structure with the ISAMD is better than that of the TMD.

6.2.2. Influence of $W_V:W_A$ on the Control Performance of the ISAMD

The timing of the "Unlock" and "Lock" switching of the ISAMD is dependent on the control law of the ISAMD and is regulated by the velocity and acceleration weight. For noncyclic near-fault and far-field ground motion loads, the shock absorption effect of the ISAMD under various weight ratios should be investigated. The results of analyzing six control performance indices of a building under control of the ISAMD with five different weight ratios and without auxiliary stiffness or a spring under excitation of 26 seismic loads are listed in Table 2. The results for the control performance indices J1–J6 are shown in Figures 16–20.

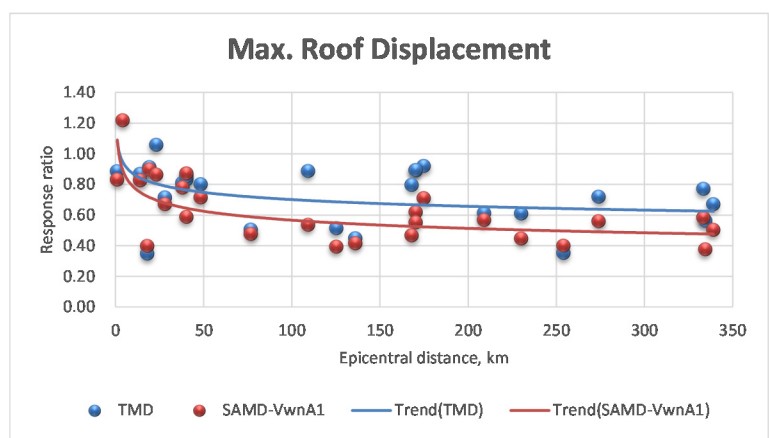

**Figure 14.** The maximal roof displacement ratio of the TMD with optimal *rf* and ISAMD $W_V : W_A = \omega_0 : 1, K = 0, C = 0$.

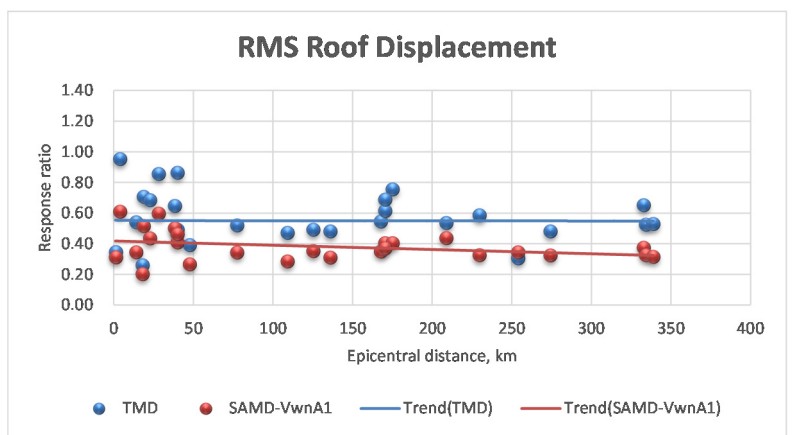

**Figure 15.** RMS roof displacement ratio of the TMD with optimal *rf* and ISAMD $W_V : W_A = \omega_0 : 1, K = 0, C = 0$.

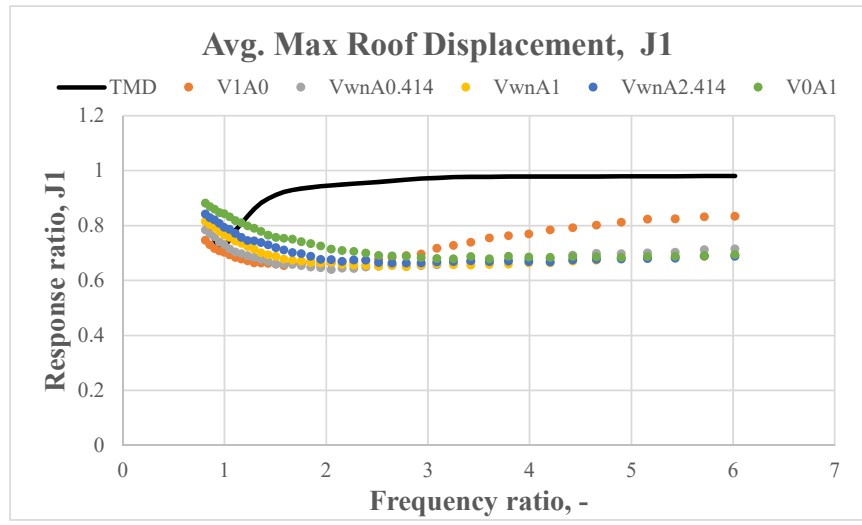

**Figure 16.** The average maximum roof displacement ratio of the building with the ISAMD, J1.

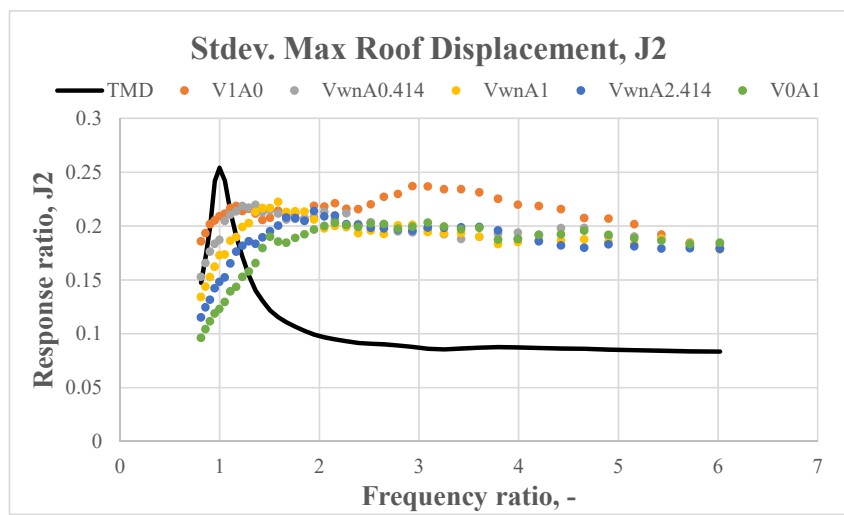

**Figure 17.** The standard deviation of the maximum roof displacement response ratios of the ISAMD, J2.

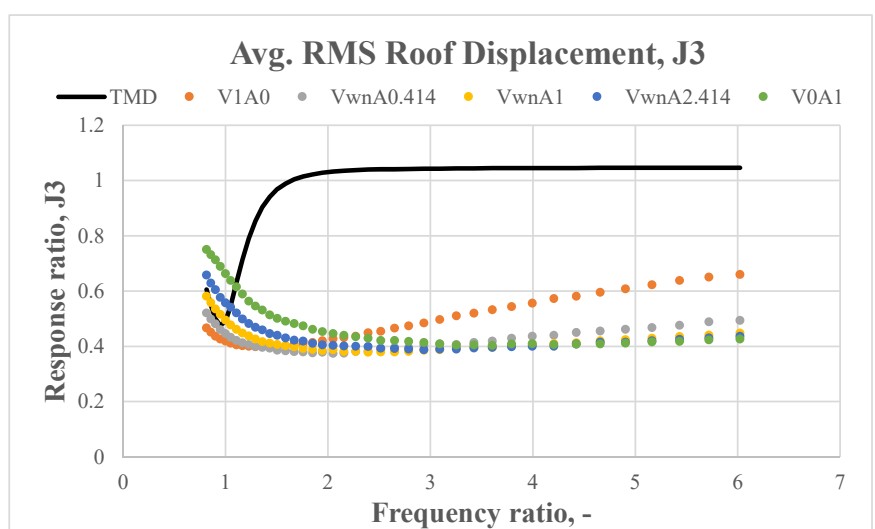

**Figure 18.** The average value of the root mean square roof displacement responses of the ISAMD, J3.

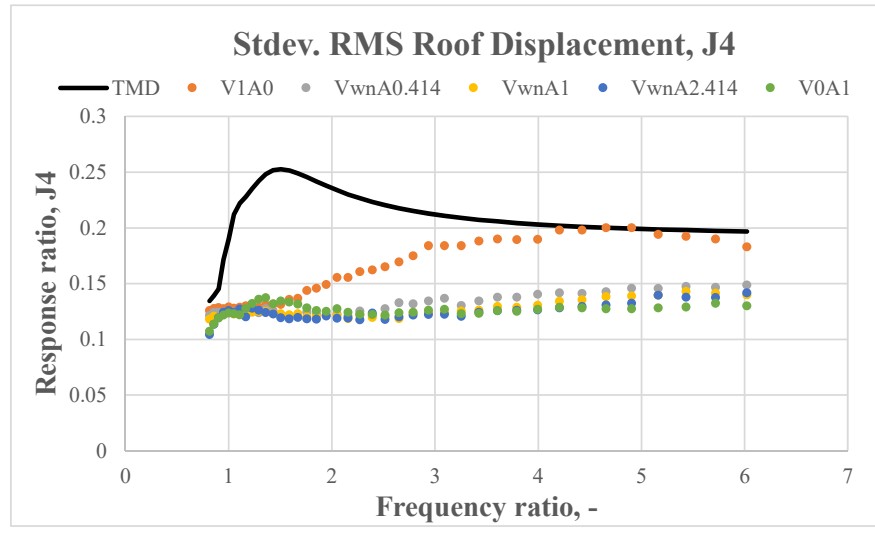

**Figure 19.** The standard deviation of the root mean square roof displacement response ratio of the ISAMD, J4.

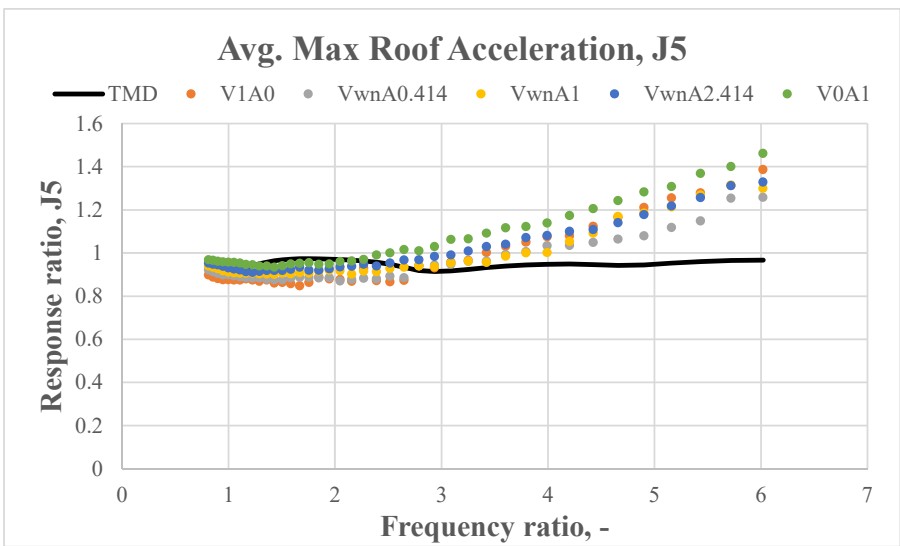

**Figure 20.** The average value of the maximum roof acceleration response ratio of the ISAMD, J5.

The minimum index of J1 was mostly within 0.65–0.67 for the ISAMD under all weight ratios, and the average seismic resistance ratios were 33–35% for the building with the ISAMD under excitation of 26 seismic records. The orange dotted line in Figure 16 shows that, when the ISAMD functions without consideration of acceleration responses (V0A1), the sensitivity of the control performance index J1 of the ISAMD to the frequency ratio ($rf$) is the highest weight ratio of the ISAMD. When the frequency ratio is greater than 2, the shock absorption effect is reduced. However, there is a large range of stiffness changes from the optimal frequency ratio, around 1.59–2. This is not a defect of design. Thus, if the weight of acceleration responses is only considered in J1, the value of J1 monotonically decreases with the frequency ratio. The frequency ratio should be very high to have the same shock absorption effect as other ratios. If it is too high, there will be other adverse reactions. Thus, adopting only the consideration of acceleration weight (V0A1) is not recommended. Figure 16 shows that the frequency ratio of 2–4 with a weight ratio of V1A0–V0A1 can achieve a stable shock absorption effect.

When the standard deviation of the maximum roof displacement reaction ratio is low, the sensitivity of the control performance to the seismic wave is low and the shock absorption control is stabler. Figure 17 shows that, when the frequency ratio is lower than 1, the value of J2 is the minimum. When the frequency ratio is large, it can steadily fall by 0.2. The shock absorption ratio is about 34% for $rf$ = 2.0. The shock absorption effect of the ISAMD can be guaranteed.

Figures 18 and 19 reveal that the root mean square roof displacement shock absorption effect of the ISAMD is better than the roof displacement shock absorption effect of the ISAMD. The average shock absorption ratio is around 60%, and the standard deviation is also relatively small. Therefore, the ISAMD provides a fine damping effect. Figure 18 shows that, for the control effect of the ISAMD, the V1A0 weight ratio is slightly inferior to the other weight ratios.

The control performance index of the ISAMD of acceleration responses behaves poorly. The acceleration reaction of the ISAMD with a frequency ratio interval of 2–4 and a better displacement control effect is basically larger than those of the bare structure. The reason is that the control force of the control mass block is close to the impact force when the frequency ratio is large. Therefore, the acceleration responses are amplified.

Another disadvantage of the ISAMD is the displacement responses of the control mass block. Figure 21 shows that the maximum displacements of the ISAMD are several times larger than those of the TMD. For example, with a frequency ratio of 2–4, the maximum displacements of the control mass block are roughly 4–6 times those of the TMD. The

large displacement responses of the control mass block restrict the application range of the ISAMD; it can only be installed in buildings with adequate space.

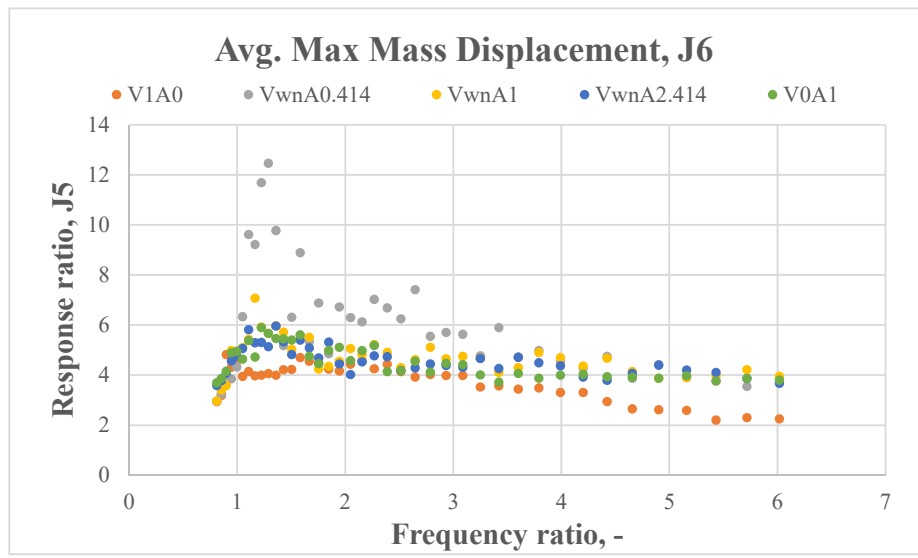

**Figure 21.** The average value of displacement responses of the control mass block of the ISAMD, J6.

### 6.2.3. Seismic Resistance and Mass Block Displacement Influence of the ISAMD with Auxiliary Damper and Spring

The large mass block displacement of the ISAMD is a major limitation of its application, as shown in Figure 21. Therefore, a feasibility study of the ISAMD with an auxiliary spring and damper installed to reduce the mass block displacement between the mass block and the structure was performed. The ISAMD in parallel with an auxiliary spring and damper is shown in Figure 22. The parameters of weight, added strength and damping coefficient interval of the ISAMD were $W_V : W_A = \omega_0 : 1$, 0, 50, 100 and 200–3600 kN/m and $c_a = 0, 20,$ 40 . . . 100 kN·s/m, respectively, to investigate the influence of seismic resistance and mass block displacement.

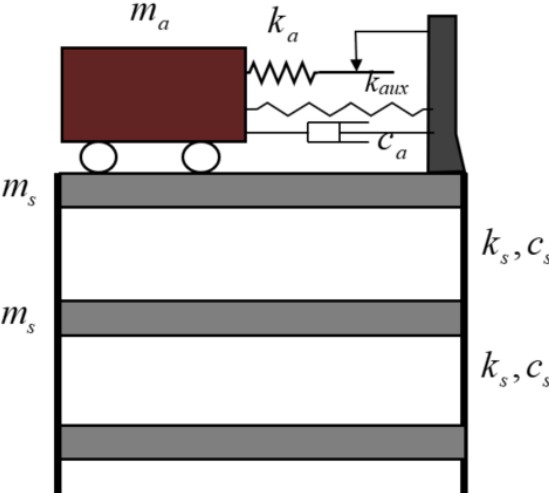

**Figure 22.** ISAMD Structure with auxiliary spring and damper.

The analysis of the control efficiency index J1 with the maximum displacement of the top floor showed that, when the stiffness of the auxiliary spring $k_{aux}$ is much lower than the optimal stiffness of the TMD, J1 decreases as $K_{aux}$ increases. When $K_{aux}$ increases up to one-quarter of the optimal stiffness of the TMD, J1 increases as $K_{aux}$ increases. Therefore, for

the analysis example in this paper, the auxiliary spring stiffness of the ISAMD is 1/8–1/4 of the optimal stiffness of the TMD to effectively reduce the mass block displacement without reducing the shock absorption effect of the ISAMD displacement control. To facilitate the comparison of the influence of an auxiliary spring and damper on the control efficiency indices J1 and J5 and the mass displacement index J6, the average value of the average efficiency indices was defined as each index of the main spring frequency ratio of the ISAMD being between 2 and 3, $W_V:W_A = \omega_0 : 1$, the better parameter interval.

$$J_k^* = \frac{\sum_{i=1}^{NCase} J_{k,i}}{NCase} \qquad (16)$$

where *NCase* is the number of analysis examples at which the frequency ratio of the ISAMD main spring is between 2 and 3.

Figures 23–25 show the influence of the auxiliary spring and damper on the top displacement control index, top floor acceleration control index and mass block displacement control index, respectively.

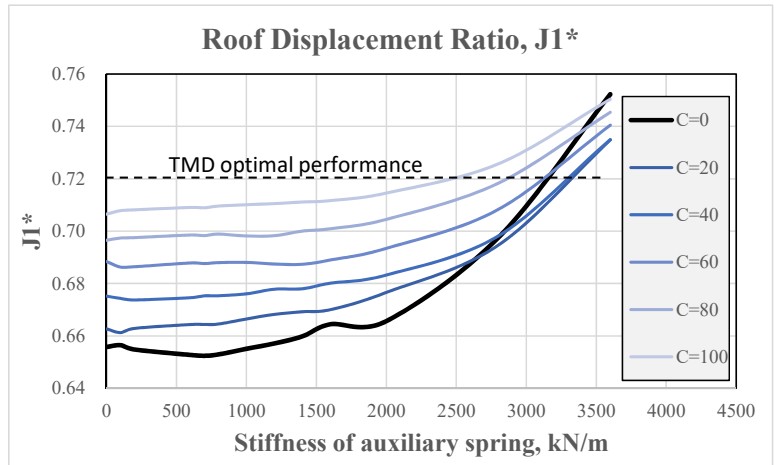

**Figure 23.** Influence of auxiliary spring and damper on the control index of maximal roof displacement.

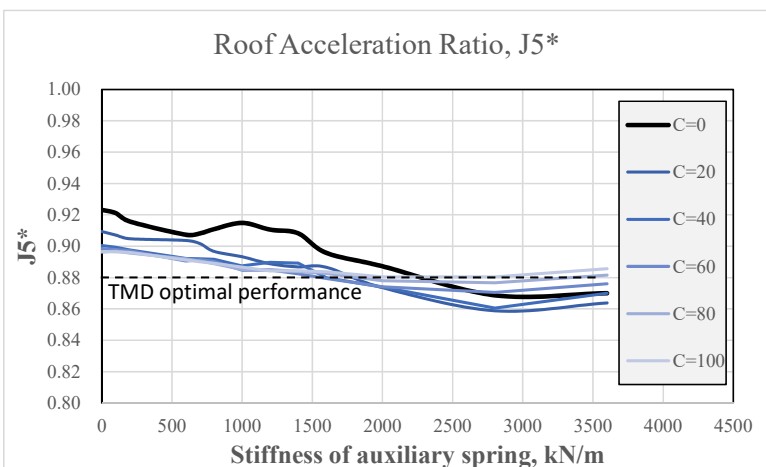

**Figure 24.** Influence of auxiliary spring and damper on control index of maximal roof acceleration.

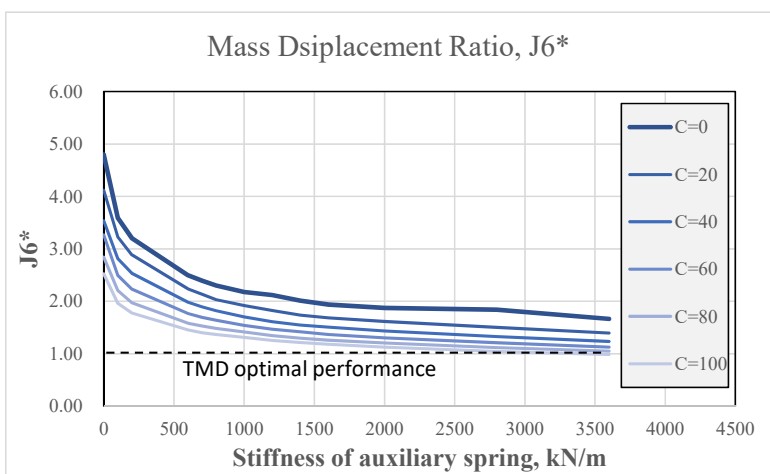

**Figure 25.** Influence of auxiliary spring and damper on control index of damper mass displacement.

The bold black lines in Figures 23–25 show the variation of the control indices of the ISAMD with only the auxiliary spring. The conclusions from the observations can be drawn as follows:

(1) Figure 25 shows that, when the spring stiffness is up to 1000 kN/m, equivalent to one-quarter of the optimal stiffness of the TMD, the mass block displacement is reduced to twice that of the TMD. The displacement of mass blocks is effectively controlled with a reduction rate of about 60% compared to the original displacement reaction.

(2) When the stiffness of the auxiliary spring is less than 1500 kN/m, the change in the displacement control effect for the maximum structural displacement reaction of the roof is less than 1%.

(3) The auxiliary spring reduces the mass block displacement without affecting the control of structural acceleration; structural acceleration at the roof is even slightly lower than that without the auxiliary spring.

(4) The ISAMD with an auxiliary damper has a more significant effect on reducing the mass block displacement than that of the ISAMD with the auxiliary spring. When the ISAMD is paired with auxiliary spring stiffness of 1000 kN/m, the damping coefficient is 80 kN·s/m, equivalent to the optimal damping coefficient of the TMD, and the mass block displacement is only 1.31 times that of the TMD. Nevertheless, the control effect index of the structural displacement reaction of the roof, J1*, is 0.70, indicating a 4% loss in seismic resistance.

(5) The ISAMD with the auxiliary damper obviously magnifies the structural displacement of the roof. The ISAMD with the auxiliary spring should be preferred to reduce the displacement of the mass block.

(6) The sensitivity of the ISAMD control indices to the frequency ratio of the main spring can be further reduced by the ISAMD with the auxiliary damper.

*6.3. Influence of Near-Fault and Far-Field Ground Motion*

The maximum structural displacement of the roof and the root mean square of the structural displacement reaction for the structure with the ISAMD under 26 seismic loads, as presented in Table 4 and Figure 26, were compared with those of the TMD. The optimal frequency ratio with the ISAMD was employed, with a weight ratio of $W_V$:$W_A$ = 0:1 and without the auxiliary spring or damper. These ground motion records were classified into two categories by the epicentral distance of 20 km. The ground motion characteristics of these seismic records and the influence of the control effect on the structure with TMD and ISAMD were compared.

**Table 4.** Comparison of optimal TMD and ISAMD ($W_V : W_A$ = 0:1, K = 0, C = 0).

| Response | TMD (max) | ISAMD (max) | TMD (rms) | ISAMD (rms) |
|---|---|---|---|---|
| Average (overall) | 0.77 | 0.63 | 0.57 | 0.38 |
| Average (<20 km) | 0.93 | 0.84 | 0.56 | 0.40 |
| Average (>20 km) | 0.73 | 0.58 | 0.58 | 0.38 |
| Stdev. (overall) | 0.26 | 0.20 | 0.17 | 0.10 |
| Stdev. (<20 km) | 0.46 | 0.29 | 0.28 | 0.16 |
| Stdev. (>20 km) | 0.18 | 0.15 | 0.14 | 0.08 |

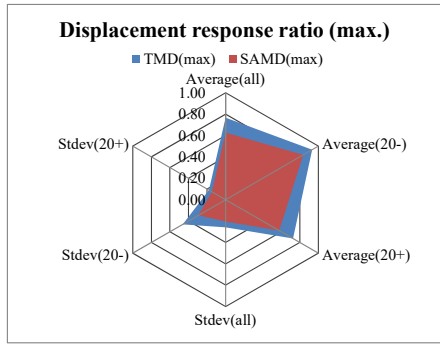 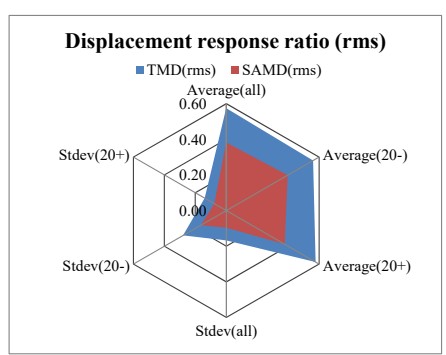

(**a**) The maximum displacement response ratio    (**b**) Root mean square (RMS) displacement response ratio

**Figure 26.** Comparison of control effects of the TMD and ISAMD.

If the epicentral distance were less than 20 km, the control effect on the structure with TMD and ISAMD would be affected, as shown in Figure 24. The average response ratios and standard deviation were higher than the ensemble average. The average maximum reaction ratio and standard deviation of the structure with the TMD were 0.93 and 0.46, respectively, indicating that the probability of the maximal roof displacement response larger than that of a structure without control is nearly 39.64%. The average maximum reaction ratio and standard deviation of the structure with the ISAMD were 0.84 and 0.20, respectively, indicating that the probability of the maximal roof displacement response larger than that of a structure without control is nearly 11.20%.

## 7. Conclusions

An ISAMD is proposed in this paper, and a directional active joint is developed as the breaker to lock and unlock contact between the structure and damper. To achieve the maximum dissipation effect, the proposed ISAMD does negative work on the structure. To investigate the seismic resistance of the proposed ISAMD, VFIFE was used to derive a mathematical model based on the characteristics of the ISAMD. Then, a 10-story shearing building controlled by the TMD or ISAMD under excitation of 26 seismic wave records was used to investigate: (1) the structural displacement reactions of the top floor and the mass block displacement reactions of the TMD and ISAMD under various combinations of parameters; (2) the displacement reduction ratio of the mass control block; (3) the influence of different parameter combinations on the seismic dissipation effect; (4) the influence of the seismic resistance and mass block displacement of the ISAMD with an auxiliary damper and spring; and (5) the influence of near-fault and far-field ground motions.

From the above analysis and results, the following conclusions are presented:

1. The maximum displacement reduction effect of the ISAMD is only slightly better than that of the TMD, reducing the maximum displacement responses of the roof by 4–10%. The optimal control efficiency indices of the TMD control system happen at a frequency ratio of 0.95. When the frequency ratio is slightly offset, the seismic

resistance of the TMD is extremely limited due to the detuning effect. The sensitivity of the control indices of the ISAMD to the frequency ratio is very low, so detuning does not occur.

2. The maximum roof displacement reaction of the structure with the ISAMD has a reduction ratio above 30%, and the root mean square of displacement reaction indicates greater than 60% seismic resistance. The control mass block displacement of the ISAMD is 2–4 times greater than that of the TMD. The installation space of the ISAMD must therefore be large. However, when the control mass frequency ratio of the ISAMD is relatively high, the installation space does not need to be excessively large.

3. The frequency ratio should be around 2–4 for the structure with the ISAMD under different weight ratios to achieve a stable shock absorption effect. The average seismic resistance ratio is about 33–35%. The shock absorption effect of the root mean square of the roof structural displacement of the structure with the ISAMD is better than that of the maximum structural displacement reaction. The average seismic proof effect is consistently around 60%. Therefore, the ISAMD provides a very good damping effect. Nevertheless, the weight ratio of the ISAMD should avoid V1A0, as the control effect is slightly inferior to those of other weights.

4. The control mass block displacement of the ISAMD can be reduced effectively by increasing the stiffness of the auxiliary spring. The gradient of the control mass block displacement on the stiffness of the auxiliary spring has a decreasing tendency with lower stiffness.

5. The seismic resistance of the TMD changes greatly with the seismic load. However, the shock absorption effect of the ISAMD is stable, and the reliability of the structural displacement control effect of the structure with the ISAMD is better than that of the TMD, regardless of earthquake distance.

6. The design criteria of the proposed ISAMD should consider a frequency ratio and weight ratio of around 2–4 and less than natural frequency, respectively.

**Author Contributions:** Conceptualization, M.-H.S and W.-P.S.; methodology, M.-H.S. and W.-P.S.; software, M.-H.S.; formal analysis, M.-H.S. and W.-P.S.; data curation, M.-H.S. and W.-P.S.; writing— original draft preparation, M.-H.S. and W.-P.S.; writing—review and editing, M.-H.S. and W.-P.S.; visualization, M.-H.S. and W.-P.S.; project administration, M.-H.S. and W.-P.S.; and funding acquisition, M.-H.S. and W.-P.S. Both authors have read and agreed to the published version of the manuscript.

**Funding:** This research was funded by Ministry of Science and Technology, Taiwan, grant number No. MOST-105-2221-E-260-003 and MOST-105-2221-M-167-001.

**Informed Consent Statement:** Not applicable for this study not involving humans.

**Data Availability Statement:** All data are available within the article and also from the corresponding author upon request.

**Conflicts of Interest:** The authors declare that there is no conflict of interests regarding the publication of this paper.

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
