# Peer review of "Seismic Resistance and Parametric Study of Building under Control of Impulsive Semi-Active Mass Damper"

_applsci, doi:10.3390/app11062468_

Round 1

Reviewer 1 Report

The paper presents a parametric study of a building where the application of an Impulsive Semi-Active Mass Damper (ISAMD) is proposed. To this scope, numerical simulations are conducted on a ten-story shearing building controlled by the TMD or ISAMD and subjected to 26 seismic-record.

The manuscript is well structured and the related results are noteworthy. Therefore, it is opinion of the Reviewer that the paper may be published after some minor revisions.

Minor comments

Firstly, the abstract should be revised just summarizing what it is reported in the paper.

The influence of fault distance on the response should be better commented (section 6.3).

Conclusions should report the main results found, highlighting the main design indications using IASMD.

Reviewer 2 Report

Please, see the attached file

Round 2

Reviewer 2 Report

It is ready to go publication !!!